# Response of Guiana dolphins to the construction of a bridge in Ilhéus, Northeastern Brazil

**Yvonnick Le Pendu**[1]*, **Alice Lima**[1], **Erica Gomes**[2], **Winnie Silva**[2], **Khamila Tondinelli Souza Cruz**[2], **Gastón Andrés Fernandez Giné**[1]

1 Department of Biological Sciences, Universidade Estadual de Santa Cruz, Campus Soane Nazaré de Andrade, Rodovia Jorge Amado, Ilhéus, Bahia, Brazil, 2 Graduate program in Tropical Aquatic Systems, Universidade Estadual de Santa Cruz, Campus Soane Nazaré de Andrade, Rodovia Jorge Amado, Ilhéus, Bahia, Brazil

* yvonnick@uesc.br

**Data Availability Statement:** Raw data are available in Biostudies repository DOI:10.6019/S-BSST1636 URL: https://www.ebi.ac.uk/biostudies/studies/S-BSST1636.

## Abstract

This study aimed to investigate whether the use of space and movements of Guiana dolphins were altered during the construction of a cable-stayed bridge at the mouth of the Cachoeira River estuary in Ilhéus, Brazil. We described and compared the visitation rate, use of space, and movements of the Guiana dolphins across two periods: before the construction began (2015–2016) and during the construction of the bridge (2017–2020). A theodolite and a total station were used to observe and record the trajectories of the dolphins. From these trajectories, we estimated the Utilization Distribution (UD) using the biased random bridge kernel (BRBK) method, the area of use, and the core area, based on the 95% and 50% BRBK density, respectively. The dolphins did not change their visitation frequency to the estuary. No significant change in area fidelity was identified by comparing the overlap of areas used during two-month periods. No change occurred in the velocity and linearity of the trajectories of the dolphins swimming close to the bridge. However, contrary to expectations, the results indicated an increase in the use of areas close to the bridge during its construction. This may have been caused by the change in the distribution of their prey in the estuary due to the emergence of a sandbank near the bridge. Bridge construction work seems not to have directly affected the Guiana dolphins' use of the area, but the impact of the construction on the local topography has changed their areas of use and core areas of activity.

## Introduction

Monitoring dolphin behavior in relation to anthropogenic activities that could potentially impact them is a useful tool for developing a better understanding of the potential adverse effects of such activities on dolphins and for planning mitigation strategies [1,2]. The construction and demolition of structures are known to negatively impact aquatic organisms [1]. Regarding dolphins, most studies focus on the impacts of vessel traffic and underwater noise

**Funding:** AL received a post-doctoral scholarship from Universidade Estadual de Santa Cruz (PROBOL-UESC), <http://www.uesc.br/arint_english/>). E G, received a scholarship grant from the Fundação de Amparo à Pesquisa do Estado da Bahia (FAPESB), <https://www.fapesb.ba.gov.br/). KC received a scholarship grant from the Coordenação de Aperfeiçoamento de Pessoal de Nível Superior (CAPES) <https://www.gov.br/capes/en>, and small grants from the Animal Behavior Society (ABS), <https://www.animalbehaviorsociety.org/web/index.php> and the Programa de Pós-Graduação em Sistemas Aquáticos Tropicais da Universidade Estadual de Santa Cruz (PPGSAT-UESC), <https://ppgsat.uesc.br/>. KC research was suported by Fundação de Amparo à Pesquisa do Estado da Bahia (FAPESB, PET0032/2012 project). W S received a scholarship grant from the Coordenação de Aperfeiçoamento de Pessoal de Nível Superior (CAPES). Funders did not play any role in the study design, data collection and analysis, decision to publish, or preparation of the manuscript.

**Competing interests:** The authors have declared that no competing interests exist.

(for example [3–5]), but few examine the effects of construction and demolition activities (for example [1,6]). Noise pollution can negatively affect marine mammals, many of which rely on sound as their primary means of exploration and communication [7,8]. Pile driving and dredging are common reasons why cetaceans temporarily avoid or stay away for long periods from areas near coastal constructions [6,9,10], but changes in water and sediment flow can also have an impact [11]. Dredging, a critical component of most major marine infrastructure developments that involves excavating and relocating sediment from lakes, rivers, estuaries, or sea beds, also affects the fish [12] that dolphins rely on to use an area [13].

Several negative impacts have been observed as a result of construction and demolition of structures, such as a decrease in the presence of resident female bottlenose dolphins (*Tursiops truncatus*) near the construction site of the John's Pass Bridge in Tampa Bay, United States [6], and an increase in the swimming speed of Indo-Pacific humpback dolphins (*Sousa chinensis*) in areas close to pile-driving activities during the construction of an aviation fuel reception tank in a bay in Hong Kong [14]. Some studies indicate that dam construction can have serious consequences for river dolphins such as the boto (*Inia geoffrensis*) [15] and Ganges river dolphins (*Platanista gangetica gangetica*) [16,17], potentially leading to unsustainable population fragmentation and even local extinction [16]. However, bottlenose dolphins in the Gulf of Mexico adapted to the impact of a bridge replacement through behavioral and temporal adjustments [9]. Similarly, Ganga River dolphins altered their preferred areas, increasing their occurrence in certain portions of the river after the construction of a barrage [18]. Bottlenose dolphins and harbour porpoises (*Phocoena phocoena*) seemed only slightly impacted by vibration piling in a Scottish estuary [19], indicating that the effects of anthropogenic activities vary according to the type of disturbance and the species concerned. However, the consequences of cumulative short-term effects of human activities are largely unknown.

Guiana dolphins (*Sotalia guianensis*) have a potentially continuous distribution in shallow coastal and estuarine waters from Nicaragua [20] to southern Brazil [21] and are the most common marine mammal in many areas exposed to anthropic development and consequent perturbations in Brazil (for example [22–25]), such as by-catch, habitat degradation, heavy vessel traffic, and exposure to pollutants. These threats have even caused a drastic decline in one of their populations in Brazil [23]. The species is classified globally as Near Threatened by the IUCN [26] and as vulnerable to extinction in Brazil [27], where it is exposed to 11 types of anthropogenic activities [28]. However, despite the species inhabiting areas affected and modified by human activities, studies on habitat use and preference have seldom considered anthropogenic factors [29,30]. To date, little is known about the impact of constructions on Guiana dolphins [28].

In Ilhéus (Northeast Brazil), a resident population of Guiana dolphins regularly visits the lower part of the Cachoeira River estuary (hereafter called Pontal Bay), mainly for foraging purposes [31]. It is the only dolphin species that frequent the estuary. As a top predator restricted to coastal waters less than 50 m deep, it plays an important role in regulating populations of fish and other marine animals of this ecosystem [32]. A cable-stayed bridge was constructed near the mouth of the estuary from September 2016 to July 2020. The construction has narrowed the mouth of the Cachoeira River, a crossing point for boats and organisms, also causing siltation in parts of the estuary. The silting process was already known before the construction of the Jorge Amado Bridge, with records of changes in bathymetry dating back to the 1970s [33]. The silting process might have intensified with the construction of the Bridge, changing the coastal physiography of Ilhéus due to the obstruction of the south-north coastal sediment drift caused by these constructions [34]. During the construction of the bridge, a small outboard boat frequently crossed the mouth of the estuary to transport materials and workers, and the installation of pillars and other construction activities caused noise pollution. It is likely that the construction of the bridge and the human activities associated with it have

impacted the Guiana dolphins in Pontal Bay, leading to behavioral changes. These changes may result from modifications in prey distribution and behavior, as well as the removal or creation of foraging opportunities due to siltation.

In this context, we aimed to assess the impact of the bridge construction on the behavior of Guiana dolphins, particularly regarding their site fidelity, space use, habitat selection, and movements. Our hypothesis is that dolphin behavior changed during the bridge construction compared to the previous period: we expected a decrease in their visitation rate to Pontal Bay, a reduction in the repeated use of sites within the bay (lower overlap of areas used), a preference for sites further away from the construction and the banks, and more linear and faster movements near the construction site.

## Methods

### Study area

We conducted this study in the lower part of the Cachoeira River estuary, popularly known as Pontal Bay, located in the municipality of Ilhéus, Bahia, Brazil (39°00′ to 3903W; 14°47′ to 14° 50S; Fig 1). This estuary is formed by the confluence of the Fundão, Cachoeira, and Santana rivers, and it flows into the Atlantic Ocean. The depth of Pontal Bay varies from 0.6 to 19.5 m and its width from 200 to 540 m [34]. It is primarily surrounded by urban areas with

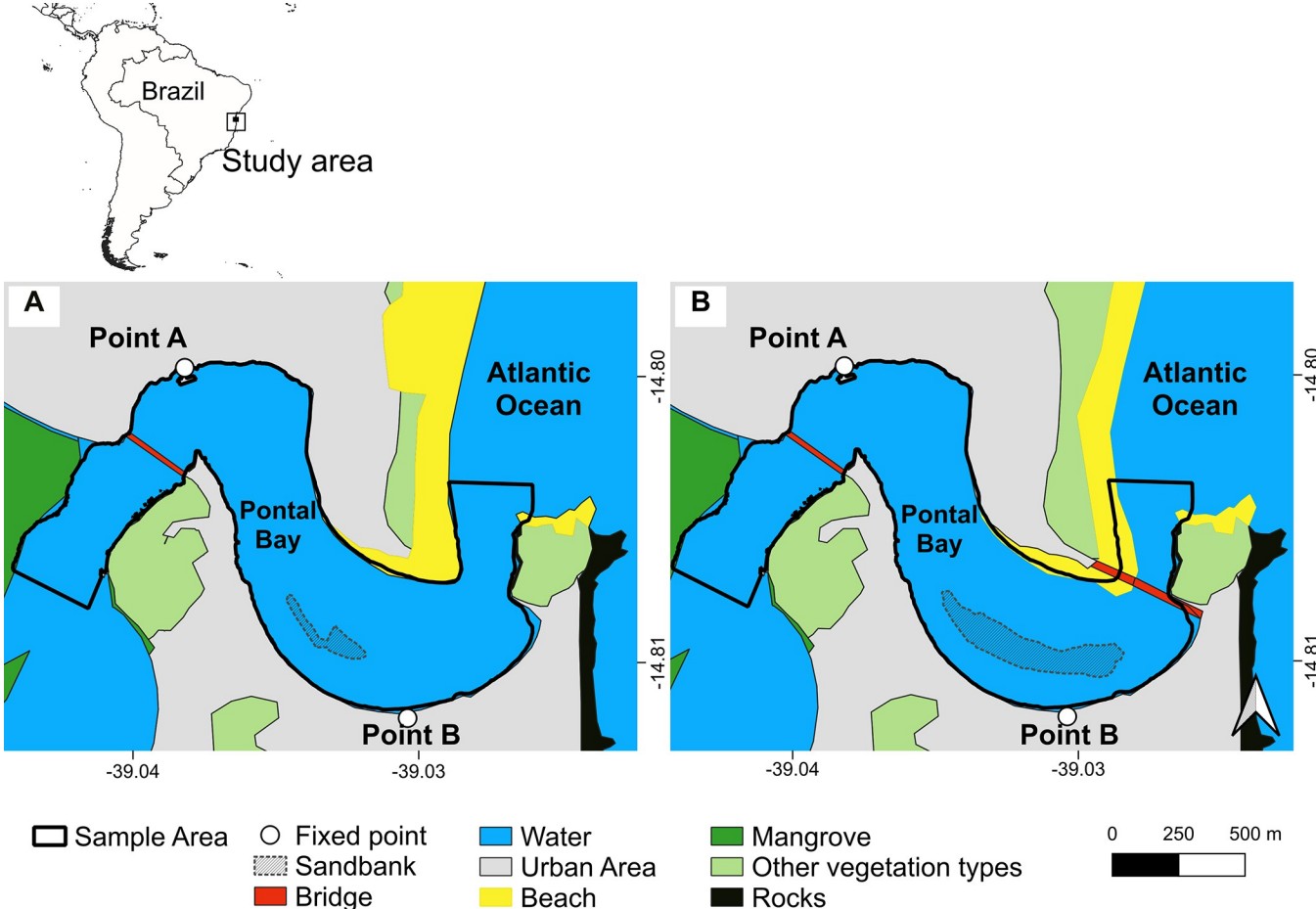

**Fig 1. Location of the study area in Ilhéus, Brazil.** Illustrative maps based on satellite images from December 2015 (A) and September 2020 (B) depicting Pontal Bay before (A) and 3 months after (B) the completion of the Jorge Amado Bridge, respectively. The Jorge Amado Bridge is shown in the eastern part of Fig 1B.

deteriorated banks where the dumping of rubble (pers. obs.) and domestic sewage is common [35,36], with mangrove vegetation present only in its western portion. The Jorge-Amado Bridge was constructed at the estuary's mouth, where the width of the bay is narrow (around 200 m). The construction of the suspension bridge began in 2016 and it was opened to traffic in July 2020. During the construction of the bridge, earthworks were carried out to facilitate the construction of the main pillar. The width of the estuary mouth was reduced, and the sandbank expanded (Fig 1). The reinforced concrete base (44 x 14 m) of the main pillar is the only submerged support remaining after the earthworks were removed at the end of the bridge construction.

## Data collection

Groups of *S. guianensis* are regularly observed off the coast of Ilhéus and in Pontal Bay [31]. Guiana dolphins can be seen all year round in groups generally of 2 to 5 individuals [31,37]. We monitored the Guiana dolphins in Pontal Bay during the period before the construction of the Jorge Amado Bridge (from April 2015 to January 2016) and during its construction (from October 2017 to September 2018 and from July 2019 to February 2020), three days a week, totaling 563.22 and 734.03 hours of sampling effort, respectively (Table **1**). During each sampling day, we conducted two monitoring sessions, each lasting 3 hours, during two of the following daytime periods: 07:00 to 10:00, 10:30 to 13:30, and 14:00 to 17:00. The observation sessions took place only when there was no precipitation. We collected data from two fixed points on the shore of Pontal Bay (Point A and Point B, Fig 1), which offered a panoramic view of the estuary. These observation points allowed us to monitor the behavior and location of Guiana dolphins across an area of 111.6 ha (sampled area), from the river mouth up to 2300 m upstream. The observation site and the two periods of the day to be monitored were pseudo-randomly selected, ensuring a similar sampling effort was maintained between the observation site and the periods.

During each observation session, we continuously scanned the sampled area with the naked eye and binoculars (Ocean Xtreme 7x50WP, Lugan brand) to detect the presence of dolphins. Once an animal or group was sighted, we used a digital theodolite (Leica Model T110) or a total station (TOPCON, ES105) with 30× magnification to triangulate the dolphins' location using two reference positions of known geographical coordinates (and elevation) to record the geographical coordinates of the dolphins [38]. A group was defined as a set of individuals swimming in an apparent association, close to each other, no more than three body lengths apart [39], moving in the same direction, and often engaged in the same activity [40]. The locations of a group were successively obtained at the shortest possible time intervals for the central position of the group. We recorded the time associated with each location, once the straight line connecting successive locations subsequently composed the group trajectories. After six minutes passed without sighting the monitored group, the successive observations of the trajectory were completed. Only trajectories comprising more than three successive locations were considered, totaling 75 trajectories recorded before the construction of the bridge and 72 trajectories recorded during the construction (Table 1). The sampling was completed prior to the COVID-19 lockdown in Ilhéus. Data collection and analysis procedures are summarized in Fig 2. No permits were required for this study, which was carried out in an unprotected area and involved the remote observation of animals in the wild.

## Data analysis

We assessed the impact of bridge construction works on the behavior of Guiana dolphins in Pontal Bay by comparing their site fidelity, space use, habitat selection, and movements before and during the construction of the bridge.

**Table 1. Sampling efforts performed before and during the construction of the Jorge Amado Bridge in Ilhéus, Brazil.**

| Sampling effort | Before the construction | During the construction | Total |
|---|---|---|---|
| Number of monitoring days | 105 | 151 | 256 |
| Number of monitoring sessions | 207 | 278 | 485 |
| Number of sessions in the site Point A | 101 | 140 | 241 |
| Number of sessions in the site Point B | 106 | 138 | 244 |
| Number of monitoring hours | 563.2 | 734.0 | 1297.3 |
| Number of registered locations | 1126 | 1332 | 2458 |
| Number of recorded trajectories | 75 | 72 | 147 |

**Impact of bridge construction on temporal and spatial site fidelity.** To assess the dolphins' temporal fidelity to the estuary, we calculated the monthly visitation rate, defined as the percentage of sessions in which dolphins entered the study area during a month. We compared the monthly visitation rates before and during the bridge construction using an independent two-sample t-test.

To assess spatial site fidelity, we used the degree of bimonthly overlap between areas of use and between core areas as indicators of spatial fidelity. We used the dataset of trajectories to estimate the Utilization Distribution (UD) of dolphins within the study area for each two-month period using the Biased Random Bridge Kernel (BRBK) method [41], implemented with the adehabitatHR package [42]. For the BRBK estimates, we used a raster resolution of 1 m, and we defined the parameter tmax (maximum time threshold for connecting two successive relocations) as 6 min, tau (interpolation time) as 3 seconds, lmin (minimum distance between successive relocations, defining intensive use or resting) as 1 m, and hmini (minimum kernel bandwidth associated with relocations) as 5 m. The D (diffusion coefficient) was estimated using maximum likelihood with the BRB.likD function from the adehabitatHR package [43]. Details on these parameters are provided in the adehabitatHR package and method description [41,42]. Considering the 95% and 50% UD probability densities, we delimited the

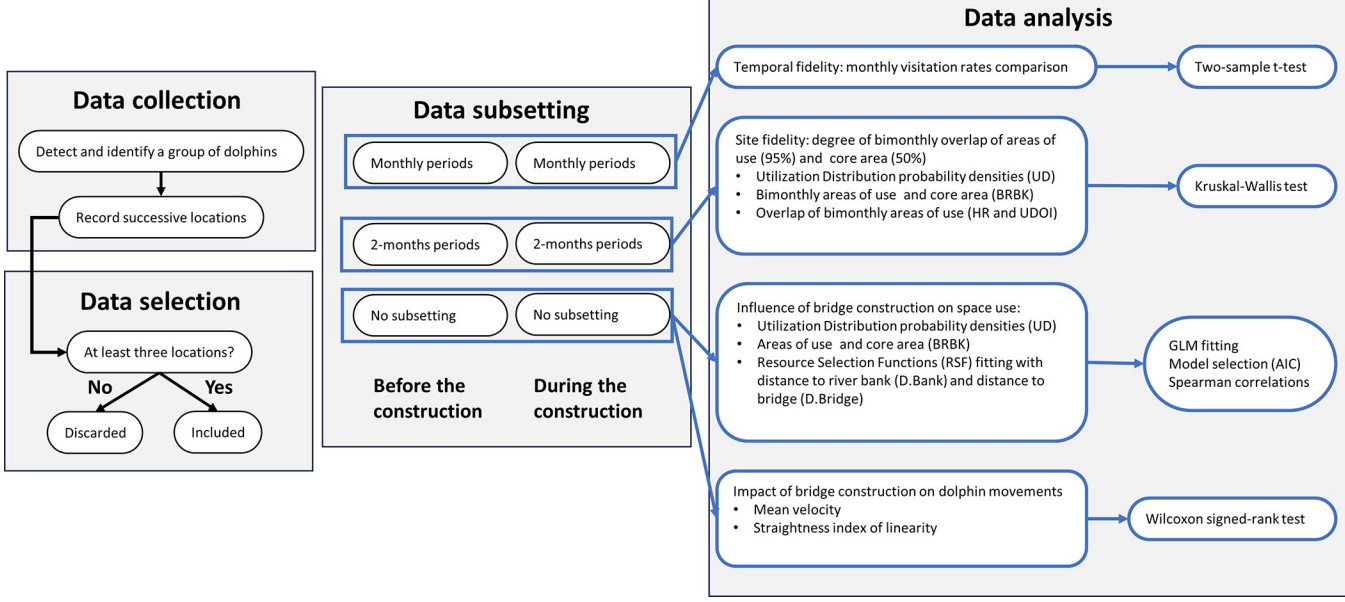

**Fig 2. Flowchart summarizing the stages of data collection, selection, organization and analysis carried out.**

bimonthly area of use (bimonthly BRBK-95%) and the core area (bimonthly BRBK-50%) of the dolphins in each two-month period. We estimated the overlap of the bimonthly areas of use (and core areas) using the Home Range Overlap Index (HR) and the Utilization Distribution Overlap Index (UDOI), considering all possible combinations of two-month periods with the kerneloverlaphr function from the adehabitatHR package [43]. Details on the overlap indices are described by [44]. We performed Kruskal-Wallis tests to evaluate differences in the overlap values estimated for each period (two treatments: before and during the bridge construction) and inter-period overlap values (comparing an additional treatment: before versus during the bridge construction) to evaluate: 1. if the site fidelity (overlap) during the bridge construction was lower than in the previous period (i.e., if there was a greater change in the space used during the bridge construction than in the previous period); and 2. if site fidelity (overlap) was lower between the two periods than within each period (i.e., if the change in used space due to the transition provoked by bridge construction was greater than expected within each period).

**Influence of bridge construction on dolphin space use and habitat selection.** We estimated the UD, area of use (BRBK-95%), and core areas (BRBK-50%) of the dolphins for the two periods separately (before and during the bridge construction) using the same BRBK method and UD probability density described above, without dividing the data bimonthly. Then, we fitted Resource Selection Functions (RSFs, *sensu* [45]) for each period to assess whether dolphins preferred to establish their areas of use and core areas further away from construction sites and riverbanks during the construction of the bridge, or if they remained as before the construction. The units selected by the animals (resource) was considered the raster cells of the bay (resolution of 1 m), and the predictor variables associated with these resource units were the distance to the river bank (D.Bank) and the distance to the bridge (D.Bridge). We systematically sampled points every 60 m in the bay to obtain values from the raster data. For each point, we extracted: the usage value, assigning 1 for points within the area of use and 0 for points outside (representing sites used and available, respectively), the distance to bridge, and the distance to the riverbank. Based on these data, we fitted GLM models with a binomial distribution and log link functions for each period separately, considering "usage" as the response variable and all combinations of the two predictor variables (D.Bank and D.Bridge). Then, we ranked all the candidate models, including the null model, based on their Akaike's Information Criterion corrected for small sample size (AICc) scores. This was done using the "dredge" function from the "MuMIn" package [46]. We selected only those models with AICc differences less than 2 ($\Delta$AICc < 2), which were considered equally plausible in explaining the observed pattern [47]. We calculated the cumulative model weights ($\Sigma\omega i$) for each predictor variable in our set of selected models, i.e., the sum of weights of all models containing the covariate [47], to evaluate the importance of the predictor variables on the dolphins' preference for establishing their area of use within the Bay before and during bridge construction. We used the sign of the estimated coefficient for each variable in the models to indicate the direction of the relationship between the predictor variables and the response (positive or negative effect). We included D.Bank and D.Bridge as predictor variables in the models because a low correlation (Spearman correlation: r < 0.5) was observed between these two variables. Additionally, we depicted the changes in the utilization distribution (UD) in Pontal Bay by calculating and plotting on a map the difference in the UD estimated during and before the bridge construction ($\Delta UD = UD_{during} - UD_{before}$), as well as estimating the Spearman correlation between the $\Delta UD$ and the predictor variables.

*Impact of bridge construction on dolphin movement (speed and linearity).* To assess whether the animals showed more linear and faster movements near the construction site during the construction period compared to before, we first selected the trajectories that occurred close to

the bridge construction (mean distance < 200 m) for each period (before and during) and calculated the mean velocity and linearity of each selected trajectory. We first estimated the following basic movement parameters: step length 'dist', the distance travelled between successive locations; net squared distance 'R2n', the squared distance between the first and last relocation of each trajectory; and the time elapsed between successive locations Δt, using the adehabitatLT package [48]. Then, we calculated the mean velocity of each trajectory, which was the average velocity of each step (dist/Δt). The linearity of each trajectory was estimated by the straightness index [49], which is calculated as the ratio of the distance between the first and last points of the trajectory to the total track length (i.e., the square root of R2n divided by the sum of the step lengths of the trajectory [50]). Finally, we used the Wilcoxon signed-rank test to compare the mean velocity and linearity of the dolphins' trajectories before and during bridge construction.

All statistics and spatial analyses were performed in R software version 4.3.2 [51]. We tested the data for normality and homoscedasticity using Shapiro-Wilk's and Levene's tests, respectively, before performing the statistical tests. Means are given ± standard deviation (SD) throughout the text.

## Results

### Temporal and spatial site fidelity

Dolphins were sighted in Pontal Bay during 18.8% (n = 91) of the 485 monitoring sessions, with a mean monthly visitation rate of 0.19± 0.09 (± SD). The monthly visitation rate did not significantly differ between the period before (0.22 ± 0.11) and during (0.16 ± 0.08) the construction of the bridge (two-sample t-test, t = 1.274; df = 21; p = 0.2168): the dolphins' fidelity of use of the estuary (here referred to as temporal site fidelity) was similar during both periods. Additionally, the dolphins' use of space within the estuary varied significantly from one bimester to another, both during and across the two study periods, with a relatively low overlap (median HR < 40%, Fig 3). There was no significant change in the degree of overlap both within and between periods for the bimonthly areas of use (BRBK-95%, Fig 3A and 3C) and core areas (BRBK-50%, Fig 3B and 3D), considering both indices: HR overlap (Kruskal–Wallis, $\chi^2$ = 1.6484, df = 2, $p$ = 0.4386; and $\chi^2$ = 1.7727, d.f. = 2, $p$ = 0.412, respectively) and the UDOI (Kruskal–Wallis, $\chi^2$ = 0.9423, df = 2, $p$ = 0.6243; and Kruskal–Wallis, $\chi^2$ = 0.7648, df = 2, p = 0.6822). These results indicate that dolphins showed no significant change in spatial site fidelity within the estuary during the bridge construction.

### Influence of bridge construction on dolphin space use and habitat selection

The estimated area used by dolphins (BRBK-95%) in the sample area was 69.0 ha before the construction of the bridge and 65.3 ha during construction, with a reduction in the area used during construction primarily in the western part, which is the part of the bay furthest from the bridge (Fig 4). The estimated core area used by dolphins (BRBK-50%) was 17.4 ha before construction and 17.5 ha during construction.

The estimated area used by dolphins (BRBK-95%) encompassed only 61.8% of the sampled area (111.6 ha) before the construction and 58.5% of the monitored area during the construction of the bridge (Fig 4). Before the bridge was built, the GLM analyses showed that the distance to the river banks (D.Bank) was the key factor influencing where dolphins chose to establish their area of use in the sampled area during the period. This variable appears in both equally plausible models (with ΔAICc < 2, Table 2), including the simpler model, and the cumulative weight of the models including this variable was 1. The effect of D.Bank was positive (Table 2), indicating that the animals preferred to use sites further away from the river

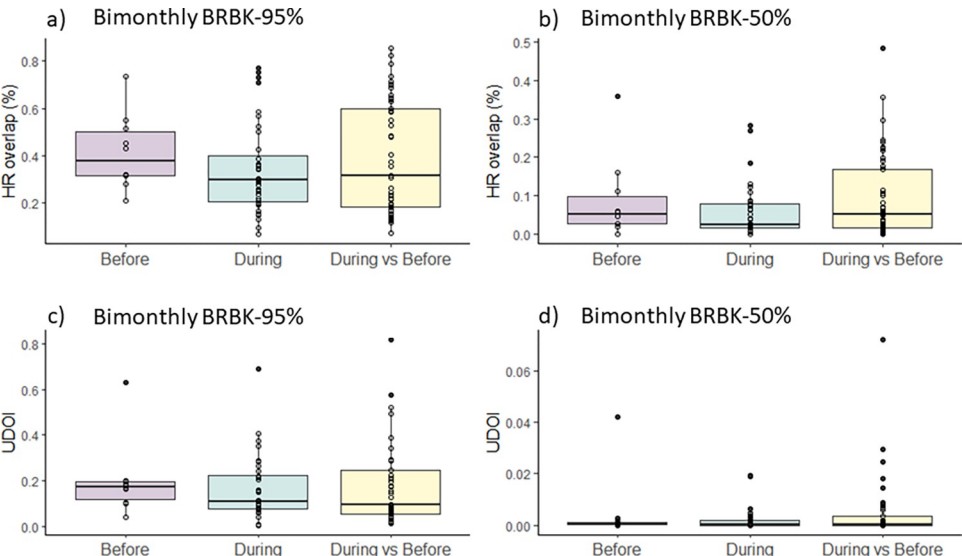

**Fig 3. Comparison of the median degree of overlap within periods (before and during the bridge construction) and between periods (before versus during the bridge construction) for the bimonthly area of use (BRBK-95%, left side) and core area (BRBK-50%, right side) of Guiana dolphins in Pontal Bay, Ilhéus, Brazil.** Top boxplots show comparisons of overlap using the Home Range overlap index (HR), while the bottom ones use the Utilization Distribution overlap index (UDOI). Within each boxplot, horizontal black bold lines correspond to median values; boxes extend from the 25th to the 75th percentile of each group's distribution of values; the whiskers (vertical lines) extend to adjacent values (i.e., the most extreme values within 1.5 interquartile range of the 25th and 75th percentile of each group); black dots beyond the whiskers denote outliers' observations (i.e., observations outside the range of adjacent values).

banks. The effect of the distance to the bridge (D.Bridge) was positive but weaker than that of the distance to the bank (D.Bank), with a cumulative weight of 0.573 (Table 2). In contrast, during the construction of the bridge, the model combining D.Bank and D.Bridge was by far the most effective in explaining dolphins' preferences: it was the only model with ΔAICc < 2 and had a high Akaike weight (ωi = 0.992; Table 2). During the construction, the effect of the D.Bank was positive, while the effect of the D.Bridge was negative (Table 2), indicating that the animals continued to prefer areas distant from the riverbanks, but began to prefer areas near the bridge during its construction. Complementarily, the difference between the UD estimated

**Table 2. Results of the AICc-based selection of GLM models conducted for Guiana dolphins (*Sotalia guianensis*) to assess their preferences before and during the construction of the Jorge Amado Bridge in Pontal Bay, Ilhéus, Brazil.** Generalized linear models used the predictor variables: distance to the bank of the river (D.Bank) and distance to the bridge (D.Bridge). We also present the number of degrees of freedom (df), differences in corrected Akaike Information Criterion (ΔAICc), Akaike weights (ωi), and the direction (positive or negative, indicated in superscript) of the effect of the predictor variables.

| Model rank | Candidate models[a] | Df | AICc | ΔAICc | ωi |
|---|---|---|---|---|---|
| a) Use before the construction of the bridge | | | | | |
| 1 | D.Bank+ + D.Bridge+ | 3 | 308.68 | 0.00 | 0.573 |
| 2 | D.Bank+ | 2 | 309.27 | 0.59 | 0.427 |
| 3 | Null model | 1 | 416.42 | 107.74 | 0.000 |
| 4 | D.Bridge- | 2 | 418.44 | 109.76 | 0.000 |
| b) Use during the construction of the bridge | | | | | |
| 1 | D.Bank+ + D.Bridge- | 3 | 352.48 | 0.00 | 0.992 |
| 2 | D.Bank+ | 2 | 362.06 | 9.58 | 0.008 |
| 3 | D.Bridge- | 2 | 404.74 | 52.26 | 0.000 |
| 4 | Null model | 1 | 422.10 | 69.62 | 0.000 |

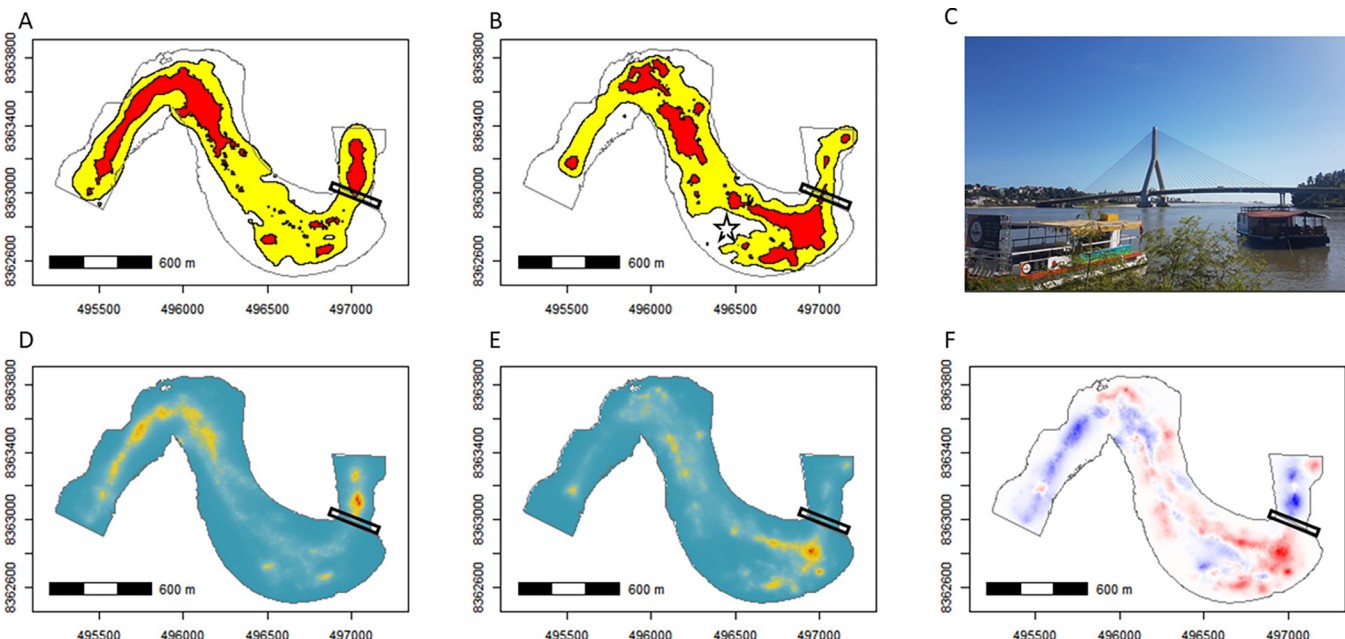

**Fig 4. Space use by Guiana dolphins (*Sotalia guianensis*) in Pontal Bay, Ilhéus, Brazil.** A and B illustrate the area of use (BRBK-95%, in yellow) and core area (BRBK-50%, in red) of the dolphins in the sampled area before and during the construction of the bridge, respectively. C shows the Jorge Amado Bridge located at the mouth of Pontal Bay (Photo: Y. Le Pendu). D and E show the spatial variation in the Utilization Distribution (UD) of the dolphins before and during the construction of the bridge, respectively (gradient from blue to red indicates the least to most used areas). F shows the difference in the estimated Utilization Distribution (ΔUD) between both periods (ΔUD = UDduring—UDbefore), where red color indicates an increase in the intensity of local use (positive values), and blue color indicates a reduction in the intensity of local use (negative values). The black and inclined rectangle in the eastern portion of the Pontal Bay maps (A, B, D, E, and F) represents the bridge, while the star in B indicates the approximate central location of the sandbank that emerged and expanded during the construction of the bridge.

before and during the construction of the bridge (ΔUD, Fig 4F) indicates an increase in the intensity of use of a site close to the bridge construction site (red color) and a reduction in the intensity of use of western sites (blue color), which are farther from the bridge.

## Impact of bridge construction on dolphin movement (speed and linearity)

There was no significant difference in the linearity (mean before = 0.3185± 0.1482, mean during = 0.2895± 0.1717) and speed (mean before = 1.4097 m/s ± 0.5578, mean during = 1.5785 m/s ± 0.7117) of dolphin movements near the bridge before and during construction (Wilcoxon signed-rank tests: linearity, t = 59, p = 0.9742; speed, t = 72, p = 0.4562).

## Discussion

In this study, we found that the use of space by Guiana dolphins in Pontal Bay changed in certain ways during the construction of the Jorge Amado Bridge, albeit not as expected. Although the site fidelity (assessed through visitation rate and bimonthly overlap area) and movement parameters (velocity and linearity near the bridge site) remained consistent before and during the construction, the dolphins concentrated their activities closer to the bridge during its construction. They reduced the use of the western portion of Pontal Bay and increased their use of an area closer to the bridge, located a few meters from it (< 200 m, Fig 4). This result is contrary to what we expected. Given that the bridge construction process entailed dredging and depositing materials (silt, sand, stones, concrete) to construct the access ramp, driving piles, a higher frequency of boats, an increased human presence, and noisy equipment along the

banks, we expected that dolphins would avoid areas close to this infrastructure project during its construction, primarily due to noise pollution and other disturbances. The lack of avoidance suggests that the Guiana dolphins have a high degree of tolerance towards such anthropogenic disturbances. If the construction of the Jorge Amado Bridge did not negatively affect the use of the area as expected, what prompted the dolphins to use an area close to the bridge during its construction? We discuss this below, considering biological aspects and physical changes close to the bridge that may have led to a change in the Guiana dolphins' preferred feeding areas in the estuary.

Guiana dolphins are considered top predators in estuarine systems [52] and primarily use the estuary for feeding [31]. Habitat selection is closely linked to foraging opportunities and habitat characteristics that improve foraging success. They prefer steeper slopes and shallow areas [30,53].

The reduction in river flow and the entry of ocean sediments have caused a sandbank to form over the last decade in the area where dolphin activity increased near the bridge and the mouth of the bay. This sandbank had already been described before bridge construction work began [34], but it has become increasingly larger and shallower since then. Ocean sediments that accumulate during high astronomical tides and northeast winds are removed from the estuary by the force of river waters [34], but the narrowing of the mouth during the construction of the bridge may have limited their removal [54]. The formation of this sandbank influenced the dolphins' area of use over time (see the location indicated by a star in Fig 4B), as it appears they avoided shallow water locations on the sandbanks, preferring areas around them instead (Fig 4B). The main reason for the increased use of this area close to the bridge during its construction is probably the new feeding opportunities that the elevation of the sandbank provided for the dolphins. It is common for Guiana dolphins to prefer areas close to breakwaters and seawalls, where they engage in "barrier-feeding", a foraging strategy in which the animals herd fish against these structures [29,55,56]. The sand bank formed in Pontal Bay may act as one of these barriers, turning the area close to it into a new and interesting foraging site. As the geomorphological and sedimentological dynamics of Pontal Bay are unpredictable, only monitoring the growth of the sandbank and the presence of dolphins will reveal whether the new opportunistic feeding area identified in this study will continue to attract dolphins or whether they will have to forage in other areas.

However, a single variable hardly explains habitat selection by dolphins [e.g. 4,32,49]. The dolphins consistently avoided locations close to the river banks before and during the construction of the bridge, suggesting that they did not use the strategy of corralling the fish on the banks of the river. The urbanized shores of Pontal Bay may pose a risk to dolphins due to the presence of construction debris and stranded boats. Under these conditions, dolphins likely choose deeper areas and only take advantage of newly formed natural barriers for foraging, as they are free from debris.

Our findings suggest that bottom depth plays a crucial role in habitat selection by dolphins, as the change in space use within this estuary by Guiana dolphins seems to be linked to changes in bathymetry. A study conducted in 2006 in Pontal bay [31] concluded that dolphins were more frequent in areas deeper than 3 m and at the end of the flood, adapting the direction of their feeding movements to the tide. Another study in the Caravelas river estuary (located approximately 400 km south of the study area) suggests that depth is the main limiting factor for habitat use by Guiana dolphins [57].

This study offers insights into the movement patterns of Guiana dolphins during significant human-induced habitat changes. The construction of bridges, involving pile driving and dredging, significantly alters the acoustic environment of dolphins. This can lead to long-term behavioral changes, potentially affecting their habitat use or causing injuries [1,28,58]. The

construction of the Jorge Amado Bridge seemingly did not have a direct impact on the behavior of the Guiana dolphins, but it did have an indirect effect on their use of space, likely due to changes in local bathymetry and the consequent redistribution of suitable feeding sites. We did not observe any short-term negative behavioral responses to the construction of the bridge, indicating a high tolerance for such disturbances. This does not mean the environment was optimal or that there were no consequences for the survival and reproduction of individuals, but rather that the dolphins can tolerate sub-optimal conditions to obtain certain resources in areas with such disturbances. We highlight that Pontal Bay has suffered continuous silting due to the deposition of continental and oceanic sediments. This is caused by the low vegetation cover on the riverbanks and the construction of port projects along the ocean coast, issues that have been neglected to date. The Jorge Amado Bridge poses yet another threat that could reduce the depth and width of Pontal Bay over time, two environmental characteristics that could decrease habitat suitability for dolphins. A continuous monitoring of Guiana dolphins in this area is highly recommended to understand the long-term impacts and to proactively conserve this population.

## Conclusions

- Guiana dolphins adjusted their use of space during bridge construction.

- Dolphins concentrated their activities close to the bridge under construction, contrary to expectations of avoidance due to worksite disturbances.

- Despite the presence of dredging, pile driving, increased boat traffic, human presence, and noise, dolphins did not avoid the bridge area, suggesting a high tolerance to some anthropogenic disturbances.

- The formation of a sandbank near the bridge influenced dolphin activity, which made greater use of the areas around the sandbank.

- The construction of the bridge indirectly affected the use of space by Guiana dolphins due to changes in local bathymetry and a probable spatial redistribution of feeding sites.

## Acknowledgments

We thank Bahia Pesca and Pontal Praia Hotel for allowing us to use their locations to observe the Guiana dolphins and the company OAS for providing the construction schedule for the Jorge Amado Bridge. We also thank Professor Niel Nascimento Teixeira and the Zoology Graduate Programme (UESC) for providing the theodolite and total station on loan and for the training in their use.

## Author Contributions

**Conceptualization:** Yvonnick Le Pendu, Erica Gomes, Winnie Silva, Khamila Tondinelli Souza Cruz.

**Data curation:** Yvonnick Le Pendu, Erica Gomes, Winnie Silva, Khamila Tondinelli Souza Cruz.

**Formal analysis:** Yvonnick Le Pendu, Alice Lima, Erica Gomes, Winnie Silva, Khamila Tondinelli Souza Cruz, Gastón Andrés Fernandez Giné.

**Funding acquisition:** Yvonnick Le Pendu, Khamila Tondinelli Souza Cruz.

**Investigation:** Erica Gomes, Winnie Silva, Khamila Tondinelli Souza Cruz, Gastón Andrés Fernandez Giné.

**Methodology:** Yvonnick Le Pendu, Erica Gomes, Khamila Tondinelli Souza Cruz, Gastón Andrés Fernandez Giné.

**Project administration:** Yvonnick Le Pendu.

**Resources:** Yvonnick Le Pendu.

**Software:** Gastón Andrés Fernandez Giné.

**Supervision:** Yvonnick Le Pendu, Gastón Andrés Fernandez Giné.

**Validation:** Yvonnick Le Pendu, Alice Lima, Gastón Andrés Fernandez Giné.

**Visualization:** Yvonnick Le Pendu, Alice Lima, Gastón Andrés Fernandez Giné.

**Writing – original draft:** Yvonnick Le Pendu, Alice Lima, Erica Gomes, Winnie Silva, Khamila Tondinelli Souza Cruz, Gastón Andrés Fernandez Giné.

**Writing – review & editing:** Yvonnick Le Pendu, Alice Lima, Gastón Andrés Fernandez Giné.

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
