## [Decision Letter · Decision Letter 0]

30 Jul 2024

PONE-D-24-14912Do bridge constructions affect dolphin use of space? Responses of Guiana dolphins from Ilhéus, Northeastern BrazilPLOS ONE

Dear Dr. Le Pendu,

Thank you for submitting your manuscript to PLOS ONE. After careful consideration, we feel that it has merit but does not fully meet PLOS ONE’s publication criteria as it currently stands. Therefore, we invite you to submit a revised version of the manuscript that addresses the points raised during the review process.

 have received of the comments of two independent reviewers. I have also critically gone through the manuscript. This is very pertinent question how the construction work affect river cetacean. As in present case "Do bridge constructions affect dolphin use of space? I find it very interesting. 

Changes in the site bathymetry due to bridge construction and the resulted sandbank formation, are the major reasons for the dolphin space use pattern. Can the Ms be correlated with habitat chief of dolphins during Covid-19 lock down i.e. in contrary to bridge construction work (disturbance) undisturbed regime also affect home range of these river cetaceans . Some such explanation about bankcard shifting of Gangetic dolphins have been provided. in following paper: COVID-19 lockdown affects zooplankton community structure in dolphin appearing site of the River Ganga at Patna_26_1

August 2023Aquatic Ecosystem Health and Management 26(1):20-31

We look forward to receiving your revised manuscript.

Kind regards,

Ram Kumar, Ph.D.

Academic Editor

PLOS ONE

4. We note that Figures 1 and 3 in your submission contain [map/satellite] images which may be copyrighted. All PLOS content is published under the Creative Commons Attribution License (CC BY 4.0), which means that the manuscript, images, and Supporting Information files will be freely available online, and any third party is permitted to access, download, copy, distribute, and use these materials in any way, even commercially, with proper attribution. For these reasons, we cannot publish previously copyrighted maps or satellite images created using proprietary data, such as Google software (Google Maps, Street View, and Earth). For more information, see our copyright guidelines: http://journals.plos.org/plosone/s/licenses-and-copyright.

a. You may seek permission from the original copyright holder of Figures 1 and 3 to publish the content specifically under the CC BY 4.0 license. 

Additional Editor Comments:

I have received of the comments of two independent reviewers. I have also critically gone through the manuscript. This is very pertinent question how the construction work affect river cetacean. As in present case "Do bridge constructions affect dolphin use of space?

Changes in the site bathymetry due to bridge construction and the resulted sandbank formation, are the major reasons for the dolphin space use pattern. Can the Ms be correlated with habitat chief of dolphins during Covid-19 lock down i.e. in contrary to bridge construction work (disturbance) undisturbed regime also affect home range of these river cetaceans . Some such explanation about bankcard shifting of Gangetic dolphins have been provided. in following paper: COVID-19 lockdown affects zooplankton community structure in dolphin appearing site of the River Ganga at Patna_26_1

August 2023Aquatic Ecosystem Health and Management 26(1):20-31

Reviewers' comments:

Reviewer's Responses to Questions

**Comments to the Author**

1. Is the manuscript technically sound, and do the data support the conclusions?

Reviewer #1: Yes

Reviewer #2: Yes

2. Has the statistical analysis been performed appropriately and rigorously? 

Reviewer #1: Yes

Reviewer #2: Yes

3. Have the authors made all data underlying the findings in their manuscript fully available?

Reviewer #1: Yes

Reviewer #2: Yes

4. Is the manuscript presented in an intelligible fashion and written in standard English?

Reviewer #1: Yes

Reviewer #2: Yes

5. Review Comments to the Author

Reviewer #1: This manuscript by Pendu et al addresses an important aspect of the impact of structural interventions on riverine species. Overall, the manuscript read smooth, figures ad tables are well organized. I recommend this manuscript may be accepted for publication. Authors may consider to incorporate the following suggestions; I hope these may help to improve the quality of the manuscript.

In introduction, please review the similar studies conducted globally. For example, author can find a similar studies on the Indus and the ganga river river dolphins. You may find Sonkar et al 2020, where they have studied the impact of large barrages on the ganga river dolphins, a similar study has been conducted by Sinha et al as well.

Methods:

Please don’t keep study area and subjects under the methodology section. You can create a separate section for study area. If possible, use some satellite imager to show the condition of river before and after the construction of bridge. It will help readers to understand the change in the river patter or morphology due to the construction activities.

In method section, if possible, include a flow chart to illustrate the overall procedure followed in this study. I believe this will help reader to quickly understand the setps followed.

Discussion:

Here you discuss your findings in chronological order and try to explain the result with finer details.

Conclusion:

Please add a separate section for conclusion. And highlight the major findings in bullet.

Reviewer #2: Manuscript entitled “Do bridge constructions affect dolphin use of space? Responses of Guiana dolphins from Ilhéus, Northeastern Brazil”. I have gone through the entire manuscript thoroughly.

This manuscript investigates the impact of the construction of the Jorge Amado Bridge on the space use and movement patterns of Guiana dolphins (Sotalia guianensis) in the Cachoeira River estuary (Pontal Bay) in Ilhéus, Brazil. The authors used theodolite and total station observations to track the dolphins' trajectories and estimate their utilization distribution, area of use, and core areas before and during the bridge construction.

The key findings of the manuscript are:

1. Contrary to the authors' expectations, the dolphins increased their use of the areas close to the bridge during its construction. This was likely due to the formation of a sandbank near the bridge, which may have provided new foraging opportunities for the dolphins.

2. The dolphins consistently avoided areas close to the river banks, both before and during the bridge construction, possibly due to the presence of debris and stranded boats.

3. The authors suggest that changes in the local bathymetry, caused by the bridge construction and the resulting sandbank formation, were the main drivers of the observed changes in the dolphins' space use.

Scientific Observations:

1. The introduction provides a comprehensive background on the potential impacts of coastal constructions on cetaceans, particularly Guiana dolphins, which are the focus of this study. The introduction is well-structured and effectively sets up the rationale for the study.

2. The methods section is detailed and the data analysis approaches are appropriate for the research questions. The use of theodolite and total station observations to track the dolphins' movements and estimate their utilization distributions is a robust approach.

3. The results are clearly presented, and the authors make good use of figures to illustrate the changes in the dolphins' space use before and during the bridge construction.

4. The discussion section provides a thoughtful interpretation of the results, considering both the direct and indirect impacts of the bridge construction on the dolphins' behavior and habitat use. The authors' hypothesis about the role of bathymetric changes in driving the observed patterns is well-supported by the data.

A few of my specific comments and suggestions are as follows:

Comment. 1

"What specific environmental changes associated with the construction of the Jorge Amado Bridge may have facilitated new foraging opportunities for Guiana dolphins, and how might these changes influence their long-term habitat selection and behavior in Pontal Bay?

Additionally, what other factors—such as prey availability, habitat complexity, and social dynamics—might influence dolphin behavior and habitat selection in response to anthropogenic disturbances?

Comment. 2

Line 42: Describe more about Guiana dolphins (Sotalia guianensis) like what functions paly in that ecosystem, How is it different from other dolphin species?, its ecological roles/ importance in Cachoeira River estuary.

Comment. 3

Line 84: Few description of under Construction Bridge should be included like, how much area was covered by the pillars of the bridge, what was the distance between the two pillars during construction? etc.

Comment. 4

Line 85: During the construction of the bridge, the width of the estuary mouth was reduced, and the sandbank expanded. This study was conducted in the estuary; have the any tidal affect observed in the behaviors of dolphin population during the study period? Or what were the visiting / sighting frequencies during the variation in the depth of the estuary.

Comment. 5

Mention the table number in the entire manuscript

Comment. 6

Justify lines and correct some minor grammatical mistakes in the manuscript

Overall, this is a well-designed and executed study that provides valuable insights into the behavioral responses of Guiana dolphins to the construction of a major coastal infrastructure project. The authors' findings contribute to the limited knowledge on the impacts of anthropogenic activities on this vulnerable species and can inform conservation and management efforts.

6. PLOS authors have the option to publish the peer review history of their article (what does this mean?). If published, this will include your full peer review and any attached files.

Reviewer #1: No

Reviewer #2: No

---

## [Author Response · Author response to Decision Letter 0]

12 Sep 2024

PONE-D-24-14912

Do bridge constructions affect dolphin use of space? Responses of Guiana dolphins from Ilhéus, Northeastern Brazil

PLOS ONE

Dear Academic editor:

First of all, we are grateful for the interest of the journal PLOS ONE' in publishing our manuscript. This revision note is a detailed response to your helpful comments and those of the two independent reviewers and complements the revised version of the manuscript, for your consideration. We have taken all comments into account, and responses to each comment are organized by reviewer.

Done

We added the following sentence in the Data collection section: “No permits were required for this study, which was carried out in an unprotected area and involved the remote observation of animals in the wild.”

3. We note that your Data Availability Statement is currently as follows: [All relevant data are within the manuscript and its Supporting Information files. Please confirm at this time whether or not your submission contains all raw data required to replicate the results of your study. Authors must share the “minimal data set” for their submission (…).

We uploaded in Biostudies repository (Licence CCO, DOI: 10.6019/S-BSST1636) the following set of 4 excel tables corresponding to the data analyzed in the article to be published in PLOS ONE: 

1.visitation rate: Visitation rate of Guiana dolphins in Pontal Bay.

2.bimonthly_overlap: Raw data used to analyze the degree of bimonthly overlap between areas of use and between core areas (HR95, HR50, UDOI95, UDOI50).

3.RSF_and_UD_diff_analysis: Resource Selection Functions and Utilization Distribution differences analysis.

4.velocity_and_linearity_analysis. Raw data used to analyze the impact of bridge construction on dolphin movement (speed and linearity) 

4. We note that Figures 1 and 3 in your submission contain [map/satellite] images which may be copyrighted. (…). 

We modified Fig 1: the map does not contain anymore original satellite image, only illustrative maps created by the authors. Fig 4 (formerly Fig 3) contains maps created by the authors, and we have mentioned the name of the author of the bridge photograph (Fig 4C) in the figure legend (Photo: Y. Le Pendu).

5. Please review your reference list to ensure that it is complete and correct. (…).

Done.

Response to academic editor comments:

Changes in the site bathymetry due to bridge construction and the resulted sandbank formation, are the major reasons for the dolphin space use pattern. Can the Ms be correlated with habitat chief of dolphins during Covid-19 lock down i.e. in contrary to bridge construction work (disturbance) undisturbed regime also affect home range of these river cetaceans. Some such explanation about bankcard shifting of Gangetic dolphins have been provided. in following paper: COVID-19 lockdown affects zooplankton community structure in dolphin appearing site of the River Ganga at Patna_26_1

August 2023Aquatic Ecosystem Health and Management 26(1):20-31

R: Although analyzing the influence of COVID-19 is certainly a promising direction for future research, our data collection was completed in February 2020, prior to the onset of the COVID-19 lockdown in our study area. The first confirmed case of the virus in Brazil was on February 25, 2020, and protective measures in Ilhéus began on March 6, which halted our sampling. Consequently, our results are not influenced by the COVID-19 lockdown, and addressing this is beyond the scope of our study. To clarify this, we have added the sentence, 'The sampling was completed prior to the COVID-19 lockdown in Ilhéus', to the Data Collection section, ensuring that readers do not misinterpret our findings.

Response to reviewer #1 comments:

This manuscript by Pendu et al addresses an important aspect of the impact of structural interventions on riverine species. Overall, the manuscript read smooth, figures ad tables are well organized. I recommend this manuscript may be accepted for publication. Authors may consider to incorporate the following suggestions; I hope these may help to improve the quality of the manuscript.

In introduction, please review the similar studies conducted globally. For example, author can find a similar studies on the Indus and the ganga river river dolphins. You may find Sonkar et al 2020, where they have studied the impact of large barrages on the ganga river dolphins, a similar study has been conducted by Sinha et al as well.

R: We added a sentence citing the review by Campbell et al. (2022) and the studies by Pavanato et al. (2016), Aggarwal et al. (2020): “Some studies indicate that dam construction can have serious consequences for river dolphins such as the boto (Inia geoffrensis)[16] and Ganges river dolphins (Platanista gangetica gangetica)[15,17], potentially leading to unsustainable population fragmentation and even local extinction [15]”. We also cite the study of Sonkar et al. (2020) in the sentence: “Similarly, Ganga River dolphins (Platanista gangetica) altered their preferred areas, increasing their occurrence in certain portions of the river after the construction of a barrage [18].”.

Methods:

Please don’t keep study area and subjects under the methodology section. You can create a separate section for study area.

R: We have renamed the "Study area and subjects" section to "Study area”. We moved the two sentences referring to dolphins [Groups of S. guianensis (…) consist of 2 to 5 individuals] to the beginning of the "Data collection" section. We hope to have contemplated your request.

If possible, use some satellite imager to show the condition of river before and after the construction of bridge. It will help readers to understand the change in the river patter or morphology due to the construction activities.

R: Figure 1 shows two illustrative maps based on satellite images depicting Pontal Bay before (A) and after (B) the completion of the Jorge Amado Bridge, in particular the evolution of the sandbank.

In method section, if possible, include a flow chart to illustrate the overall procedure followed in this study. I believe this will help reader to quickly understand the setps followed.

R: We added a new figure 2, a flowchart summarizing the stages of data collection, selection, organization and analysis carried out.

Discussion: Here you discuss your findings in chronological order and try to explain the result with finer details.

Correct

Conclusion: Please add a separate section for conclusion. And highlight the major findings in bullet.

R: A separate section for conclusion was added, with the major findings highlighted in bullets, although such conclusion in bullets does not seem to be a common practice in articles published in PLOS ONE. According to guidelines, “Results, Discussion and Conclusions may all be separate, or may be combined to create a mixed Results/Discussion section (commonly labeled “Results and Discussion”) or a mixed Discussion/Conclusions section (commonly labeled “Discussion”)”.

Response to reviewer #2 comments: 

Comment. 1. "What specific environmental changes associated with the construction of the Jorge Amado Bridge may have facilitated new foraging opportunities for Guiana dolphins, and how might these changes influence their long-term habitat selection and behavior in Pontal Bay?

R: Thank you for this suggestion. We pointed out in the third paragraph of the discussion [The reduction in river flow...] that the formation of a sandbank near the bridge was the environmental change that facilitated new foraging opportunities, as it probably functions as a barrier that allows fish to be trapped. At the end of the third paragraph of the discussion, we have added the following sentence to address the long-term impact of environmental changes: “As the geomorphological and sedimentological dynamics of Pontal Bay are unpredictable, only monitoring the growth of the sandbank and the presence of dolphins will reveal whether the new opportunistic feeding area identified in this study will continue to attract dolphins or whether they will have to forage in other areas.”. Potential long-term effects are outlined at the end of the discussion: “We highlight that Pontal Bay has suffered continuous silting due to the deposition of continental and oceanic sediments.” (…) “The Jorge Amado Bridge poses yet another threat that could reduce the depth and width of Pontal Bay over time, two environmental characteristics that could decrease habitat suitability for dolphins. A continuous monitoring of Guiana dolphins in this area is highly recommended to understand the long-term impacts and to proactively conserve this population.”

Additionally, what other factors—such as prey availability, habitat complexity, and social dynamics—might influence dolphin behavior and habitat selection in response to anthropogenic disturbances?

R: The multiplication of anthropogenic disturbances along the coastal zone can have an impact on dolphin behavior. However, the manuscript focuses on the impacts of bridge construction on the space use of dolphins in Pontal Bay, which is only part of the home range of these dolphins, that also feeds and socializes in the coastal waters of the ocean. 

Comment. 2. Line 42: Describe more about Guiana dolphins (Sotalia guianensis) like what functions paly in that ecosystem, How is it different from other dolphin species?, its ecological roles/ importance in Cachoeira River estuary.

R: We added the following information to the text: (…) Guiana dolphins regularly visits (… Pontal Bay), mainly for foraging purposes [31]. It is the only dolphin species that frequents the estuary. As a top predator restricted to coastal waters less than 50 m deep, it plays an important role in regulating populations of fish and other marine animals of this ecosystem [32]. 

Comment. 3. Line 84: Few description of under Construction Bridge should be included like, how much area was covered by the pillars of the bridge, what was the distance between the two pillars during construction? etc.

R: We added the following sentences: “During the construction of the bridge, earthworks were carried out to facilitate the construction of the main pillar. The width of the estuary mouth was reduced, and the sandbank expanded (Fig 1). The reinforced concrete base (44 x 14 m) of the main pillar is the only submerged support remaining after the earthworks were removed at the end of the bridge construction.”

Comment. 4. Line 85: During the construction of the bridge, the width of the estuary mouth was reduced, and the sandbank expanded. This study was conducted in the estuary; have the any tidal affect observed in the behaviors of dolphin population during the study period? Or what were the visiting / sighting frequencies during the variation in the depth of the estuary.

R: We did not analyze the effect of tides in this study but added information on this topic in the discussion: “A study conducted in 2006 in Pontal bay [31] concluded that dolphins were more frequent in areas deeper than 3 m and at the end of the flood, adapting the direction of their feeding movements to the tide. Another study in the Caravelas river estuary (located approximately 400 km south of the study area) suggests that depth is the main limiting factor for habitat use by Guiana dolphins [57]”. 

Comment. 5 Mention the table number in the entire manuscript

R: “Table 2” is now quoted in the text along with the numerical results.

Comment. 6. Justify lines and correct some minor grammatical mistakes in the manuscript

We have made a second in-depth revision to avoid formatting and grammatical errors.

Response to reviewer #3 comments: 

Few comments:

It appears that Guiana dolphins forage in shallow water and estuaries and therefore, sand banks created due to bridge construction and other barriers facilitated herding of fishes by dolphin towards the barrier. It is an opportunistic foraging behavoiur. However, it imperative to look at the long-term impact of such barriers and adopted behaviour on the dolphin. 

Authors would like to add such prospectives in their introduction as well as discussion. This will help in formulating long term planning for systematic monitoring of such projects which affects the critical aquatic lives.

R: Thank you for your comments and suggestions. We have added several sentences to address the long-term impact of environmental change in the introduction “These changes may result from modifications in prey distribution and behavior, as well as the removal or creation of foraging opportunities due to siltation.” and in the discussion: “As the geomorphological and sedimentological dynamics of Pontal Bay are unpredictable, only monitoring the growth of the sandbank and the presence of dolphins will reveal whether the new opportunistic feeding area identified in this study will continue to attract dolphins or whether they will have to forage in other areas.”. Potential long-term effects are outlined at the end of the discussion: “We highlight that Pontal Bay has suffered continuous silting due to the deposition of continental and oceanic sediments.” (…) “The Jorge Amado Bridge poses yet another threat that could reduce the depth and width of Pontal Bay over time, two environmental characteristics that could decrease habitat suitability for dolphins. A continuous monitoring of Guiana dolphins in this area is highly recommended to understand the long-term impacts and to proactively conserve this population.”

---

## [Editor Report · Decision Letter 1]

27 Sep 2024

PONE-D-24-14912R1Do bridge constructions affect dolphin use of space? Responses of Guiana dolphins from Ilhéus, Northeastern BrazilPLOS ONE

Dear Dr. Le Pendu,

Thank you for considering all the comments and suggestions made by the reviewers. I have critically perused the manuscript and I find that the Ms still need some minor revision to strengthen the results obtained in the Ms.  After careful consideration, Therefore, we invite you to submit a revised version of the manuscript that addresses the following points: .

Do bridge constructions affect dolphin use of space? Responses of Guiana dolphins from Ilhéus, Northeastern Brazil

I have critically perused the manuscript .Title: Actually I am not in favour of a title in Interrogation; Interrogation is more often used for direct manipulation/experimental study

However for space use by dolphins there are several determinants including river water, morphometric flow patterns etc. Authors themselves explains in the text as “However, a single variable hardly explains habitat selection by dolphins [e.g. 4, 32,49].” Thus I would suggest a better title that could reflect this kind of study as follow:

“Effect of bridge constructions on dolphin use of space: A case study of Guiana dolphins from Ilhéus, Northeastern Brazil'

Or

"Responses of Guiana dolphins from Ilhéus Northeastern Brazil to bridge constructions"

Regarding the comments belowComments for text:  

I think earlier statement has been misconstrued  i.e. Can the Ms be correlated with habitat shift of dolphins during Covid-19 lock down i.e. in contrary to bridge construction work (disturbance) undisturbed regime also affect home range of these river cetaceans.

I did not intend to mention the effect of Covid-19; as other studies have reported reduced human disturbance due to covid -19 lock down

Some such explanation about bankward shifting of Gangetic dolphins have been provided in following paper: COVID-19 lockdown affects zooplankton community structure in dolphin appearing site of the River Ganga at Patna_26_1 August 2023Aquatic Ecosystem Health and Management 26(1):20-31

The comment does not suggest mentioning Covid -19 effects. Covid-19 did not directly affect any nonhuman animal, however the lockdown offered opportunity to study behavioural changes in wild lives in the absence of human activities. So the comment suggests Habitat shifting, space use/ food base etc. in relation to anthropogenic disturbances. I am still of the view that this will further strengthen the quality of the paper. Covid-19 lock down was considered as reduced anthropogenic pressure /activity /disturbance from river bank side and no motor boat activity, undoubtedly these are determinants of shifting of dolphin habitat.

The suggested paper has (i) presented data on food base of dolphin and habitat shifting as this manuscript directly explains the new opportunistic feeding area attracting dolphins. (ii) a probable spatial redistribution of feeding sites in lack of human activities during the lock down .

It may be kept in mind that the Covid -19 lockdown is just a means that facilitated negligible or lower human disturbance. Justifiably anthropogenic disturbance are the direct determinant not the Covid-19. Please submit your revised manuscript by Nov 11 2024 11:59PM. If you will need more time than this to complete your revisions, please reply to this message or contact the journal office at plosone@plos.org. Please include the following items when submitting your revised manuscript:A rebuttal letter that responds to each point raised by the academic editor and reviewer(s). You should upload this letter as a separate file labeled 'Response to Reviewers'.A marked-up copy of your manuscript that highlights changes made to the original version. You should upload this as a separate file labeled 'Revised Manuscript with Track Changes'.An unmarked version of your revised paper without tracked changes. You should upload this as a separate file labeled 'Manuscript'.If applicable, we recommend that you deposit your laboratory protocols in protocols.io to enhance the reproducibility of your results. Protocols.io assigns your protocol its own identifier (DOI) so that it can be cited independently in the future. For instructions see: https://journals.plos.org/plosone/s/submission-guidelines#loc-laboratory-protocols. Additionally, PLOS ONE offers an option for publishing peer-reviewed Lab Protocol articles, which describe protocols hosted on protocols.io. Read more information on sharing protocols at https://plos.org/protocols?utm_medium=editorial-email&utm_source=authorletters&utm_campaign=protocols.

We look forward to receiving your revised manuscript.

Kind regards,

Ram Kumar, Ph.D.

Academic Editor

PLOS ONE

Journal Requirements:

Additional Editor Comments:

Do bridge constructions affect dolphin use of space? Responses of Guiana dolphins from Ilhéus, Northeastern Brazil

I have critically perused the manuscript . Actually I am not in favour of a title in Interrogation; Interrogation is more often used for direct manipulation/experimental study

Howeevr for space use by dolphins there are several determinants including river water, morphometric flow patterns etc. Authors themselves explains in the text as “However, a single variable hardly explains habitat selection by dolphins [e.g. 4, 32,49].” Thus I would suggest a better title that could reflect this kind of study as follow:

“Effect of bridge constructions on dolphin use of space: A case study of Guiana dolphins from Ilhéus, Northeastern Brazil'

Or

"Responses of Guiana dolphins from Ilhéus Northeastern Brazil to bridge constructions"

Regarding the comments below

Authors misconstrued my comments i.e. Can the Ms be correlated with habitat shift of dolphins during Covid-19 lock down i.e. in contrary to bridge construction work (disturbance) undisturbed regime also affect home range of these river cetaceans.

I did not intend to mention the effect of Covid-19; as other studies have reported reduced human disturbance due to covid -19 lock down

Some such explanation about bankward shifting of Gangetic dolphins have been provided in following paper: COVID-19 lockdown affects zooplankton community structure in dolphin appearing site of the River Ganga at Patna_26_1 August 2023Aquatic Ecosystem Health and Management 26(1):20-31

The comment does not suggest mentioning Covid -19 effects. Covid-19 did not directly affect any nonhuman animal, however the lockdown offered opportunity to study behavioural changes in wild lives in the absence of human activities. So the comment suggests Habitat shifting, space use/ food base etc. in relation to anthropogenic disturbances. I am still of the view that this will further strengthen the quality of the paper. Covid-19 lock down was considered as reduced anthropogenic pressure /activity /disturbance from river bank side and no motor boat activity, undoubtedly these are determinants of shifting of dolphin habitat.

The suggested paper has (i) presented data on food base of dolphin and habitat shifting as this manuscript directly explains the new opportunistic feeding area attracting dolphins. (ii) a probable spatial redistribution of feeding sites in lack of human activities during the lock down .

It may be kept in mind that the Covid -19 lockdown is just a means that facilitated negligible or lower human disturbance. Justifiably anthropogenic disturbance are the direct determinant not the Covid-19.

---

## [Author Response · Author response to Decision Letter 1]

2 Oct 2024

Response to reviewers

PONE-D-24-14912

Dear Ram Kumar:

We are grateful for the interest of the journal PLOS ONE' in publishing our manuscript. This revision note is a response to your two comments on the revised version of our manuscript, for your consideration. 

Title: Actually I am not in favour of a title in Interrogation; Interrogation is more often used for direct manipulation/experimental study 

However for space use by dolphins there are several determinants including river water, morphometric flow patterns etc. Authors themselves explains in the text as “However, a single variable hardly explains habitat selection by dolphins [e.g. 4, 32,49].” Thus I would suggest a better title that could reflect this kind of study as follow:

“Effect of bridge constructions on dolphin use of space: A case study of Guiana dolphins from Ilhéus, Northeastern Brazil' Or "Responses of Guiana dolphins from Ilhéus Northeastern Brazil to bridge constructions"

As suggested, we have adopted an affirmative form for the title: Response of Guiana dolphins to the construction of a bridge in Ilhéus, Northeastern Brazil.

I think earlier statement has been misconstrued i.e. Can the Ms be correlated with habitat shift of dolphins during Covid-19 lock down i.e. in contrary to bridge construction work (disturbance) undisturbed regime also affect home range of these river cetaceans. I did not intend to mention the effect of Covid-19; as other studies have reported reduced human disturbance due to covid -19 lock down Some such explanation about bankward shifting of Gangetic dolphins have been provided in following paper: COVID-19 lockdown affects zooplankton community structure in dolphin appearing site of the River Ganga at Patna_26_1 August 2023Aquatic Ecosystem Health and Management 26(1):20-31 The comment does not suggest mentioning Covid -19 effects. Covid-19 did not directly affect any nonhuman animal, however the lockdown offered opportunity to study behavioural changes in wild lives in the absence of human activities. So the comment suggests Habitat shifting, space use/ food base etc. in relation to anthropogenic disturbances. I am still of the view that this will further strengthen the quality of the paper. Covid-19 lock down was considered as reduced anthropogenic pressure /activity /disturbance from river bank side and no motor boat activity, undoubtedly these are determinants of shifting of dolphin habitat. The suggested paper has (i) presented data on food base of dolphin and habitat shifting as this manuscript directly explains the new opportunistic feeding area attracting dolphins. (ii) a probable spatial redistribution of feeding sites in lack of human activities during the lock down . It may be kept in mind that the Covid -19 lockdown is just a means that facilitated negligible or lower human disturbance. Justifiably anthropogenic disturbance are the direct determinant not the Covid-19.

We understood that your previous comment referred to the decrease in anthropogenic pressure resulting from the lockdown and not to any direct effects of COVID. Although analysing the influence of COVID-19 lockdown is certainly a promising direction for future research, our results are not influenced by the COVID-19 lockdown, because 'The sampling was completed prior to the COVID-19 lockdown in Ilhéus'. We added this sentence to the Data Collection section in the R1 revised version of the manuscript and detail the chronology of the events in our previous response to reviewers.

Consequently, we cannot comment about “a probable spatial redistribution of feeding sites in lack of human activities during the lock down” or any other change in the dolphins’ behaviour during the lockdown because we did not conduct visual monitoring of the dolphins in the area during the lockdown. It was forbidden precisely because of the confinement requirement.

---

## [Editor Report · Decision Letter 2]

8 Oct 2024

Response of Guiana dolphins to the construction of a bridge in Ilhéus, Northeastern Brazil

PONE-D-24-14912R2

Dear Yvonnick Le Pendu,,

We’re pleased to inform you that your manuscript has been judged scientifically suitable for publication and will be formally accepted for publication once it meets all outstanding technical requirements.

Kind regards,

Ram Kumar, Ph.D. D. Sc (H/C) 

Academic Editor

PLOS ONE

---

## [Editor Report · Acceptance letter]

6 Nov 2024

PONE-D-24-14912R2 

PLOS ONE

Dear Dr. Le Pendu, 

I'm pleased to inform you that your manuscript has been deemed suitable for publication in PLOS ONE. Congratulations! Your manuscript is now being handed over to our production team.

Kind regards, 

on behalf of

Professor Ram Kumar 

Academic Editor

PLOS ONE